# Towards GPU-driven Code Execution for Distributed Deep Learning

Changho Hwang*    KyoungSoo Park    Ran Shu    Xinyuan Qu†    Peng Cheng    Yongqiang Xiong

*KAIST*    *KAIST*    *Microsoft Research*    *UCAS*    *Microsoft Research*    *Microsoft Research*

*Abstract*—Modern state-of-the-art deep learning (DL) applications tend to scale out to a large number of parallel GPUs. Unfortunately, we observe that the collective communication overhead across GPUs is often the key limiting factor of performance for distributed DL. It under-utilizes the networking bandwidth by frequent transfers of small data chunks, which also incurs a substantial I/O overhead on GPU that interferes with computation on GPU. The root cause lies in the inefficiency of CPU-based communication event handling as well as the inability to control the GPU's internal DMA engine with GPU threads.

To address the problem, we propose a *GPU-driven* code execution system that leverages a GPU-controlled hardware DMA engine for I/O offloading. Our custom DMA engine pipelines multiple DMA requests to support efficient small data transfer while it eliminates the I/O overhead on GPU cores. Unlike existing GPU DMA engines initiated only by CPU, we let GPU threads to directly control DMA operations, which leads to a highly efficient system where GPUs drive their own execution flow and handle communication events autonomously without CPU intervention. Our prototype DMA engine achieves a line-rate from a message size as small as 8KB (3.87x better throughput) with only 4.32μs of communication latency (9.1x faster) while it incurs little interference with computation on GPU, achieving 1.82x higher all-reduce throughput in a real training workload.

## I. Introduction

Modern machine learning (ML) applications tend to harness an increasingly larger number of accelerators (especially GPUs in this work) [7], [10]. State-of-the-art deep learning (DL) algorithms often need to scale out to thousands of GPUs for higher throughput and accuracy [10]. Unfortunately, this poses a substantial communication overhead to the entire system, which harms GPU utilization by delaying or interfering with numeric computation.

The communication overhead mainly arises in two different aspects. First, collective communication (e.g., all-reduce, split-and-gather, all-to-all, etc.), which is widely adopted in most of popular DL algorithms, often splits sending data into multiple small chunks for pipelining or to send to multiple different destinations. The chunk size tends to get smaller as we scale out, which is detrimental to efficient utilization of networking bandwidth. Second, popular communication libraries for GPUs such as NCCL [16] and RCCL [3] often incur a severe I/O overhead on GPU. This is because they commonly leverage memory-mapped I/O (MMIO) for data copies between GPUs, which consumes a substantial amount of GPU resources (i.e.,

core cycles and L2 cache/DRAM bandwidth). We observe that concurrent execution of collective communication and numeric computation on GPU interferes with each other, which often drops the parallel computation throughput by 45.0% while achieving only 53.6% of the peak communication throughput in our real training application (BERT-Large [4], see details in Section II-C).

Existing systems tackle only one of the two issues as it is difficult to handle both at the same time. For example, we can avoid the I/O overhead by offloading the I/O to a hardware DMA engine instead of MMIO using GPU threads. The DMA engine is already implemented on commodity GPUs, but unfortunately, it is initiated only by CPU threads. Thus, leveraging the DMA engine enrolls CPU on the critical path of communication. This incurs the CPU-GPU synchronization overhead that increases communication latency, especially detrimental to the throughput of small data chunk transfer. For example, existing implementations that leverage the DMA engine in popular DL frameworks often incur hundreds of μs of communication latency. Similarly, if we do not leverage the DMA engine for low-latency communication with small data chunks, we sacrifice the I/O overhead on GPU, instead.

This paper proposes *GPU-driven* system, a communication-motivated DL system design. The key idea of GPU-driven system is autonomous execution control of GPU code without any control by external devices. This regime tightly connects computational power of every GPU core across the entire cluster by allowing GPU threads to communicate directly with remote GPUs without any external control signals, obtaining low-latency communication. At the same time, to avoid the I/O overhead on GPU, we design a GPU-controlled DMA engine. Specifically, our custom DMA engine is designed to be directly initiated by GPU threads for I/O offloading, which avoids the heavy MMIO without CPU intervention.

Our evaluation shows that our DMA engine prototype is especially beneficial for small messages, achieving a high communication throughput (3.87x over `cudaMemcpy` with 8KB messages) at low latency (9.1x faster over CPU intervention). Furthermore, it does not interfere with computation on GPU, which delivers both computation and communication throughput gains over using MMIO-based communication libraries (1.82x faster all-reduce in BERT-Large [4] training, see Section IV-B). We expect that our design would help a variety of distributed DL applications, including data-, tensor-, and expert-parallelism.

---
*Now at Microsoft Research.
†This work was done during the internship program in Microsoft Research.

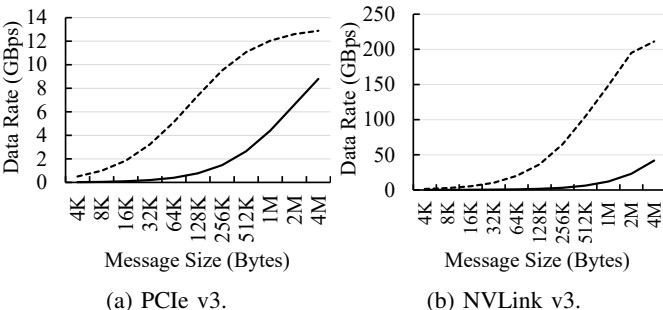

(a) PCIe v3.    (b) NVLink v3.

Fig. 1: Data dependency between GPUs decreases the inter-GPU data rate due to event handling delays. Solid lines refer to actual data rate (for sending one message at a time) in TensorFlow's CPU-controlled communication crossing a PCIe v3 or a NVLink v3 switches while dashed lines indicate the ideal data rate without event handling delays.

## II. BACKGROUND & MOTIVATION

### A. Small Data Transfer in Collective Communication

Collective communication consists of several communication primitives that concurrently exchange the data across multiple GPUs, which is widely adopted to implement various parallelism methods in distributed DL. Popular use cases include *all-reduce* for data-parallelism, *split-and-gather* for tensor-parallelism [9], [23], and *all-to-all* for expert-parallelism [5]. As the number of employed GPUs gets larger, the size of unit data transfer in collective communication becomes smaller as it splits the local data into multiple pieces to be delivered to different GPUs. This small transfer size makes the overall performance of collective communication highly dependent on the control plane overhead before and after each data transfer. Unfortunately, we observe that the control plane overhead either with *CPU-controlled* or even *GPU-controlled* communication is pretty substantial (See Section II-B and Section II-C). Also, existing workarounds (e.g., *tensor fusion* [22]) that batch a large amount of data to avoid small transfers would not completely address the problem as they trade off computational throughput by intentionally delaying data transfer.

### B. External Execution Control Overhead

Existing GPU program execution heavily relies on an external processor (i.e., CPU) to submit GPU commands for kernel execution or data transfer. Unfortunately, this model often incurs a large overhead due to the delay for command delivery from the host side to GPU hardware queue (i.e., *stream*). One can use the conventional GPU event interface (i.e., cudaEvent) to hide the delay, but it would also suffer from fairly substantial delay incurred by event handling. When adopted to inter-GPU communication (unlike NCCL [16]), which we call *CPU-controlled* communication, we observe that event handling becomes the primary cause for large communication delay rather than the data transfer itself.

We consider a common communication scenario where two GPUs have a data dependency – one GPU receives computa-

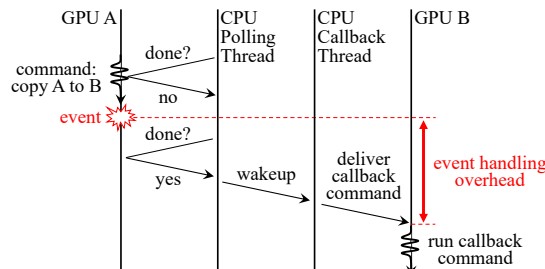

Fig. 2: CPU intervention in inter-GPU event handling.

| Overhead Detail | Delay (μs) |
|---|---|
| Initiation | |
| Trigger send ready event on the GPU | 3.8 |
| Sync comp. stream and comm. stream | 11.6 |
| Completion Check | |
| Event polling gap | 58.3 |
| Delay of pthread mutex lock | 58.7 |
| GPU kernel launch overhead | 19.2 |
| **Total** | **151.6** |

TABLE I: Breakdown of the constant overhead of inter-GPU data transfer using TensorFlow in Fig. 1.

tion results of another GPU to use them as inputs of its own computation. In every data transfer, event handling is needed to check the dependency between the copy and the GPU commands around the copy operation, which degrades the actual data rate between GPUs. Fig. 1 compares the ideal inter-GPU data rate (cudaMemcpy throughput) with the actual data rate in TensorFlow's CPU-controlled communication, which is still used along with NCCL especially for model-parallelism implementations. We see that the event handling overhead with cudaMemcpy drastically lowers the data rate both in the PCIe and NVLink interfaces. We explain two event handling scenarios when GPU A sends data to GPU B.

*1) Runtime Intervention for the Control:* CPU can serve as an intermediary to deliver an event between two communicating GPUs. In fact, if GPUs are located in different NUMA nodes or on different machines, the runtime intervention by CPU is required for communication. Also, some frameworks like TensorFlow implement a generic interface that always uses CPU for GPU event handling regardless of the placement. Fig. 2 illustrates the event handling overhead due to CPU intervention when GPU A sends its data to GPU B that plans to run the next command with the data.

We notice three places for the overhead. First, it is inefficient for a CPU thread to poll GPU events because the event interface disallows the CPU thread to monitor multiple events at the same time. While it takes only ∼3μs for a dedicated busy-waiting CPU thread to be notified of a triggered GPU event,[1] this approach does not scale when an application has to run many parallel tasks, which will run many polling threads. Instead, the event polling loop of TensorFlow uses only one CPU thread, which incurs a ∼58.3μs of polling gap on average (see Table I). Second, it takes time to wake up the CPU thread

---

[1]Please refer to the experiment setup in Section IV.

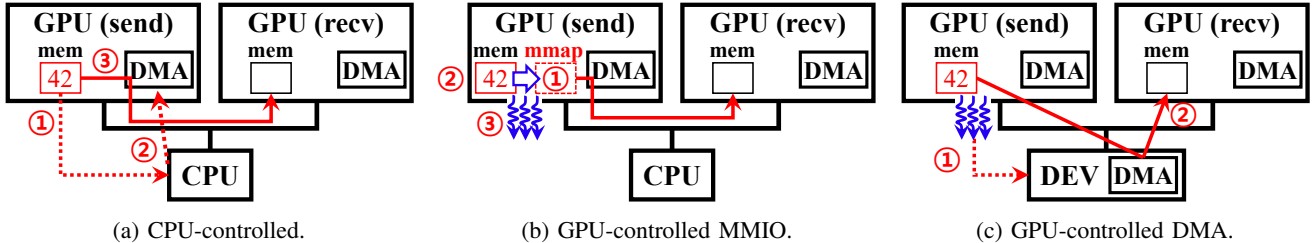

(a) CPU-controlled.  (b) GPU-controlled MMIO.  (c) GPU-controlled DMA.

Fig. 3: Comparison between CPU-controlled and GPU-controlled communication – the latter has two different approaches, which leverage (b) MMIO (like NCCL) or (c) directly initiated DMA (this work). DEV refers to any kinds of devices that can implement our DMA engine.

that invokes the callback function of the triggered event. In TensorFlow, it takes ∼58.7μs for the callback thread to acquire the mutex lock from when it is released by the polling thread. This delay could be reduced to as low as 5μs if both threads are running on the same CPU core, but co-locating the threads or even merging them into a single one would increase the event polling interval as well as the overall processing time. Lastly, it is inefficient for the callback thread to deliver the computation command to GPU B. Delivering the event signal to GPU B would take only 2∼3μs if implemented efficiently,[2] but we need to deliver the callback command binary as well. This extra delay could be avoided if we can deliver the GPU command ahead of time and trigger it later on the CPU side, but this is not supported by commodity GPU.

*2) Asynchronous Control:* If the GPUs are under the same NUMA node, CPU can reserve a GPU event to be triggered asynchronously so that GPUs can directly communicate with each other when the event occurs. In this case, one can deliver the callback command to GPU B before the actual event and use the conventional GPU event interface (i.e. `cudaEvent` or a higher-level wrapper such as CUDA Graphs [11]) to trigger the callback command on GPU B with GPU A's event. Ideally, this should take as short as sending a single bit from GPU A to GPU B. However, we find that triggering a GPU event (∼4μs) and waking up a dependent GPU command (10∼20μs) are disappointingly slow – it ends up taking as much as sending the command binary to the GPU at runtime. We suspect that this is due to inefficient hardware implementation on GPU for event handling. In TensorFlow, this overhead contributes to the delay for initiating a transfer that depends on GPU computation as shown in Table I.

### C. I/O Overhead of GPU-controlled Communication

Since CPU intervention incurs a large overhead, how about managing the communication with GPU itself? NCCL [16] [3] leverages *GPUDirect* [17] to enable this approach, which exposes the GPU memory space for peer-to-peer access so

that GPU threads can read/write data to/from another GPU.[4] As GPU threads can directly invoke data copy, they can handle communication events efficiently without the involvement of CPU. Since commodity GPU hardware does not allow GPU threads to initiate its own DMA engine, GPU-controlled communication leverages MMIO, which will implicitly conduct DMA when GPU threads write data on the mapping. Fig. 3 compares CPU-controlled and GPU-controlled communication. The former (Fig. 3a) takes the following steps: ① CPU is notified when data is ready, ② CPU initiates the DMA engine, and ③ DMA copies the data. On the other hand, GPU-controlled communication with MMIO (Fig. 3b) follows ① CPU creates a memmory map (mmap) of the destination GPU's address space prior to the runtime execution, ② data is ready at runtime, and ③ GPU threads copy the data into the mmap'ed region, which implicitly conducts DMA copy.

Unfortunately, data copying by GPU threads often heavily interferes with parallel kernel computation, especially due to L2 cache pollution and warp scheduler operations. Specifically, a data-copy GPU thread needs to load the data onto its register file for data transfer, but this pollutes the L2 cache when reading DRAM as bypassing the L2 cache is not supported by commodity GPUs [20]. It leads to severe performance degradation over initiating DMA directly, as the latter copies the data on DRAM directly to the I/O bus (PCIe or NVLink). Additionally, the copying threads frequently issue load/store instructions that often make warp schedulers busy, which makes other threads for parallel computation yield their clock cycles. Although the affected computation threads are limited to those that co-run warp schedulers with data-copy threads, they delay the entire kernel by falling behind the other threads.

To analyze the impact of the contention, we measure the slowdown of two different GPU kernels that heavily access only a specific type of GPU resources each: L2 cache (1.96 TBps read) and warp schedulers (2.02 IPC),[5] respectively (all numbers measured on a V100 GPU), while

---

[2]This is roughly estimated based on that it takes ∼2μs for a GPU thread to read a 4-byte data on the host DRAM and it takes ∼3μs for a busy-waiting CPU to read a GPU event.

[3]Equally applied to RCCL [3] on AMD GPU as well. For convenience, we borrow the terms from CUDA or NVIDIA GPUs, which can be easily converted into corresponding terms in OpenCL or AMD GPUs.

[4]Actually, CPU-controlled communication also leverage GPUDirect to conduct `cudaMemcpy` between peer GPUs efficiently without crossing the root complex, but its execution path is different from that of GPU-controlled communication.

[5]Heavy usage of warp schedulers means frequent instruction fetches, i.e. large instructions per cycle (IPC). > 99.2% of instructions are FFMA.

running concurrently with NCCL (v2.11.4) 64 MB all-gather[6] kernels using 8x V100 GPUs. We leverage NVIDIA Visual Profiler (NVVP) and Nsight Compute to verify that (1) the L2 cache kernel shows near-zero DRAM access and L1 data cache hit rate and (2) the warp schedulers kernel shows near-zero L2 cache/DRAM throughput. We have also verified the concurrency of computation and all-gather kernels and no other CPU/GPU activities during the experiment. In this experiment, the slowdowns due to L2 cache and warp schedulers contention are up to 2.35x and 2.00x, respectively, where it slows down either the computation or the concurrent NCCL communication (when one side is degraded less, the other side tends to be impacted more). This result shows that heavy contention could arise depending on the GPU resource usage of concurrent computation kernels.

We run a microbenchmark to evaluate the contention of NCCL all-reduce during data-parallel training of a BERT-Large [4] model. This model performs 32 MB of all-reduce at a time, which issues 4 MB data transfer in parallel with eight GPU workers. On a server with 8x V100 GPUs (connected with a single PCIe switch (16x PCIe v3)), the parallel computation throughput drops by 45.0% while all-reduce achieves only 5.0 GBps on average, degraded to 53.6% of the peak throughput without the interference. On a server with 8x A100 GPUs (connected with a NVSwitch (NVLink v3)), the slowdown of all-reduce is even worse – the parallel computation throughput drops by 14.3% while the NCCL all-reduce operation achieves only 30.9% of the peak throughput (49.0 GBps).

## III. DESIGN & PROTOTYPE IMPLEMENTATION

This section presents our approach with the *GPU-driven* code execution that avoids the communication overhead on GPU without CPU intervention.

### A. GPU-controlled DMA Engine

We claim that a GPU-controlled DMA engine (Fig. 3c) can eliminate the communication overhead, which in turn serves as the basis of our GPU-driven system. The GPU-controlled DMA engine enables a GPU thread to directly initiate DMA operations when the data is ready (①), which will immediately push the data into the I/O bus without wasting GPU cycles (②). While this would deliver low-latency communication without the MMIO overhead, it is non-trivial to realize this feature. In fact, an ideal implementation would be to modify the existing DMA engine on GPU to support GPU-controlled DMA, but it is infeasible as we cannot update the GPU hardware.

Instead, we consider employing an external device as illustrated in Fig. 3c at the cost of extra communication latency from GPU threads. We pursue a general DMA engine design that can be implemented as either software or hardware on any hardware platforms (e.g., CPU, GPU, SmartNIC, FPGA, etc.) or I/O bus types (PCIe, NVLink [19], or Infinity Fabric Link

(xGMI) [2]). Regardless of the platform, all implementations need to share the same runtime interface for GPU kernels. Also, the DMA interface should support low latency and flexibility while meeting the different requirements of software and hardware engines. One key issue lies in the design of a DMA request message from GPU, which we call a *send request* (SR), as it has significant impact on the performance and the implementation complexity.

In terms of hardware, receiving a large SR whose size exceeds the data bus width (64 bits in modern 64-bit processors) will take multiple cycles, which would require SR buffer management, reassembly of segmented SRs, and handling dropped SRs (caused by SR buffer overflow). As implementing them on hardware would significantly complicate the logic and increase the spatial cost, we share an 8-byte SR design for both software and hardware engines. While it is challenging to hold the metadata of a general memory copy (two addresses and a copy length) within 8 bytes, we address this by adopting a small number of send/recv buffers, which reduces the address space by replacing general 8-byte addresses with a few bits of buffer indices. This is feasible thanks to the static nature of collective communication where the communicating entities are fixed – it enables offline pre-scheduling of data transfers so that receivers know which data arrives at which buffer without any additional metadata received at runtime. Meanwhile, the DMA requests on different buffers are pipelined for low latency and high throughput.

In terms of software, keeping an SR buffer would be more efficient as it would otherwise require extra control to prevent overwriting a previous SR. That is, unlike a hardware implementation where a fully received SR can immediately trigger the internal DMA pipeline at every cycle, a software thread could overwrite an unread SR unless the sender (GPU) coordinates with the receiver (the DMA stack) prior to sending a new SR. Unfortunately, such coordination would incur an extra delay as the GPU needs to read a remote flag on the DMA stack before sending an SR. We address this issue by maintaing a specialized ring buffer for SR, where the GPU checks only a local replica of the buffer head before sending an SR, and the replica is asynchronously updated by the DMA stack. This removes the coordination delay from the critical path of communication while providing a consistent SR interface for both software and hardware engines.

### B. DMA Engine Prototype Implementations

We first present our software DMA engine that harnesses CPU as the data plane while GPU serves as the control plane. Then, we introduce our hardware DMA engine that allows us to glimpse at the high performance achieved by an ideal implementation. Both software and hardware engines are designed for cross-machine communication, while current prototype implementations support only intra-machine communication.

*1) Software Engine:* We implement a CPU thread that busy-waits for SRs and invokes `cudaMemcpy` accordingly, i.e., it leverages the existing hardware DMA engine on the sender GPU. Note that this is different from CPU-controlled

---

[6]We use all-gather as it only performs communication without any extra computation such as reduction in all-reduce.

| Module Name | ALMs | | BRAM Blocks | |
|---|---|---|---|---|
| | # | Capacity | # | Capacitry |
| FPGA Stack | 14253 | 3.34% | 188 | 6.93% |
| PCIe | 1364 | 0.32% | 13 | 0.48% |

TABLE II: Resource usage of a single hardware DMA stack.

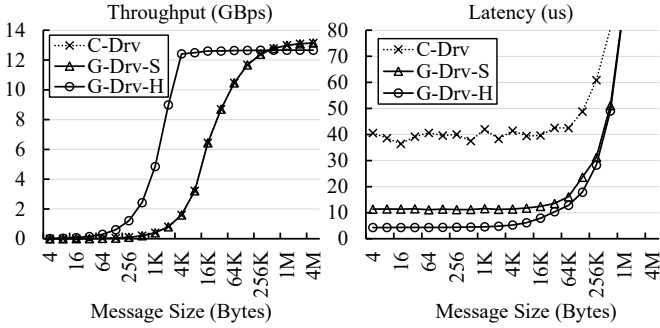

Fig. 4: Performance comparison between the CPU-controlled communication (C-Drv) and the GPU-controlled DMA engines (G-Drv-S (software) and G-Drv-H (hardware)).

communication as we use CPU only for data plane operations while the control plane (event handling) is managed by GPU threads. For high throughput, the busy-waiting loop drains all SRs in the ring buffer and invoke `cudaMemcpy` once for sending on a continuous memory space. Also, instead of slow `cudaEvent`, we use MMIO for the CPU-GPU communication that delivers SR, SC (Send Completion), and RC (Receive Completion) signals, which takes only 2∼3µs.

Alternatively, the software engine can perform MMIO with CPU threads instead of initiating the hardware DMA engine, which can reduce the `cudaMemcpy` overhead (i.e., sending a copy request from CPU to the DMA engine on GPU). However, this approach fails to achieve the line rate in most host CPU architectures due to their poor throughput of crossing the PCIe root complex [24], [25]. This issue might be resolved in the future CPU architectures or by leveraging ARM cores on SmartNICs [26], which is left as our future work.

*2) Hardware Engine:* We implement a custom hardware with FPGA for DMA operations, which delivers two benefits over our software engine prototype. First, our hardware engine avoids the extra communication delay incurred by the overhead of `cudaMemcpy` as it performs DMA directly. Second, unlike existing hardware DMA engines on GPU, our custom hardware implements pipelining of multiple parallel DMA operations. This helps achieve a high data rate even for sending small data chunks. Table II shows resource usage of our implementation on an Intel Arria 10 FPGA.

Note that our FPGA prototype is limited to support the communication between only two GPUs and it does not support NVLink as there is no programmable hardware (or an off-the-shelf device) that can connect to NVLink. Instead, we consider it as a proof-of-concept that demonstrates the ideal benefit rather than a practical device that can be deployed on a large scale. A more practical implementation would be realized by future advances in CPU, GPU, or SmartNICs.

*C. GPU-driven System Design*

GPU-controlled DMA engines would be easily adopted by existing systems, e.g., NCCL can replace its MMIO with initiating our DMA engines. However, existing systems would not fully exploit the benefit of GPU-controlled communication as the communication APIs are launched by CPU – the CPU intervention barrier still remains between computation and communication. We propose a GPU-driven system design that removes this barrier and tightly connects computation and communication by running the entire DL application in a single kernel, called a *loop kernel*. Our key observation is that online dynamic scheduling is unnecessary as DL workloads are typically deterministic at runtime. Unlike existing systems

that dynamically launch GPU kernels using CPU at runtime, our GPU-driven system automatically merges all kernels into a loop kernel (one for each GPU) at compile time and launches it when the application starts, which will repeatedly run the application during the entire lifetime. A loop kernel is generated by a code generator that reads an operational graph of a DL application and automatically assembles corresponding code snippets of GPU operators to build loop kernel code.

GPU-driven system design lets GPUs fully control the application, which would minimize the event handling overhead for inter-GPU communication. While it often delivers the computation-side benefit as well by enabling finer-grained GPU core scheduling inside the loop kernel, we leave it as our future work and focus on the communication-side benefits in this paper.

IV. PRELIMINARY EVALUATION

**Hardware.** We use an Intel Xeon Gold 5118 CPU (24 lcores, 2.30 GHz) and eight NVIDIA V100 GPUs connected through a single PCIe v3.0 switch. Our hardware DMA engine is implemented on an Intel Arria 10 FPGA.

**Software.** We evaluate our system using CUDA 11.3, and all baselines for comparison use their own latest recommended versions of software dependencies at the time of writing.

*A. DMA Engine Performance*

Fig. 4 compares the performance of communication between two GPUs with our prototype DMA engine over a CPU-controlled communication baseline. We measure the throughput by sending many parallel messages at the same time and reporting the maximum throughput achieved with varying message sizes. For latency measurements, we implement a ping-pong application and report one-way latency – unlike throughput measurements, this includes communication event handling delays. This experiment assumes a favorable scenario for the CPU-controlled baseline where we can adopt the asynchronous control (explained in Section II-B2). In this scenario, a one-way trip requires triggering only two GPU events and two stream synchronizations.

In the left graph of Fig. 4, our software engine (G-Drv-S) shows the same throughput as that of C-Drv, since both use

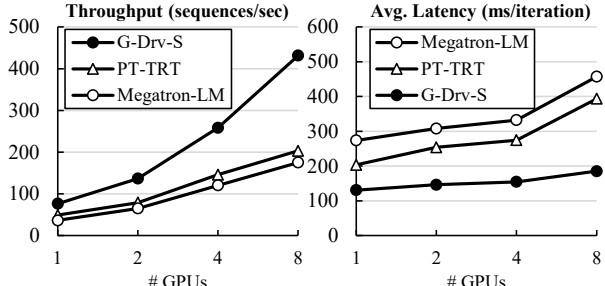

Fig. 5: BERT-Large data-parallel training throughput and average latency per iteration with varying numbers of GPUs (sequence length 384, batch size 10, mixed-precision).

cudaMemcpy for the data-plane. In contrast, our hardware engine (G-Drv-H) shows a huge throughput improvement, saturating the bandwidth with only 8 KB messages while G-Drv-S needs 4 MB messages for saturation. This is because the hardware DMA engine pipelines processing multiple DMA requests while cudaMemcpy cannot. This improvement would be especially beneficial when GPU sends multiple messages to different destinations at the same time, e.g., all-to-all communication for expert-parallelism, which is popular for scaling out state-of-the-art Transformer-based models [5].

We note that the maximum achieved throughput of G-Drv-H is 3.68% lower than G-Drv-S. This is because an external DMA stack needs to send both read and write requests to sender and receiver GPUs, respectively, while the native DMA engine on the sender GPU needs to send only write requests. However, as the gap is small, it would not affect the end-to-end application performance much.

The right graph of Fig. 4 shows that the one-way latency of C-Drv is at least ~39.3μs on average. In contrast, G-Drv-S and G-Drv-H achieve 3.5x and 9.1x better latency, respectively. This is because our DMA engines handle the communication events directly in GPU threads while C-Drv relies on the cudaEvent interface that suffers from large overhead to trigger the events and synchronize streams. This improvement would be especially beneficial when GPUs perform split-and-gather of intermediate results to distribute the workload, as in tensor-parallelism [9], [10]. One thing to note about our DMA engine is that the benefit is obtained with little GPU cycle consumption. We evaluate this in the following section.

### B. Computation-Communication Interference Evaluation

To compare the interference between computation and communication of using NCCL against using our DMA engine, we evaluate data-parallel training throughput of G-Drv-S by training one of the representative NLP models, BERT-Large [4], with two comparison baselines using NCCL: Megatron-LM [10] (PyTorch [1]-based framework with NCCL) and PT-TRT (PyTorch+TensorRT [18]+NCCL). For baseline experiments, we use the official BERT-Large implementations from the official public repositories [13], [15] without code modification, and leverage their provided Docker container

images without modification except that we upgrade NCCL into version 2.11.4. We use TensorRT version 7.1.2.

Fig. 5 shows that G-Drv-S outperforms Megatron-LM and PT-TRT by 2.46x and 2.12x with 8 GPUs, respectively. While G-Drv-S delivers computational gains from our GPU-driven system design (Section III-C), we observe that a larger gain comes from benefits of using our DMA engine. Specifically, 64.5% of the end-to-end gap between G-Drv-S and PT-TRT with 8 GPUs is obtained as NCCL operations slow down due to the interference of MMIO with back-propagation computation, showing only 5.0 GBps of all-reduce throughput. On the other hand, our DMA engine suffers near-zero interference by initiating DMA directly instead of using MMIO, achieving 9.10 GBps of all-reduce throughput (1.82x faster).

## V. DISCUSSION & FUTURE WORK & RELATED WORK

We expect that hardware advances in near future would enable more efficient implementations. For example, implementing our software DMA engine on SmartNIC would avoid the throughput issue of the PCIe root complex [25] via direct PCIe connection with GPUs (e.g., NVIDIA H100 CNX [14] combines GPU with SmartNIC), which enables efficient MMIO on SmartNIC. NVIDIA announces that they are willing to put hardware accelerators for inter-GPU communication on SmartNICs (e.g., all-to-all engine on NVIDIA BlueField-3 [12]), which implies that a similar implementation with our hardware engine might be realized in the future. Additionally, host CPU architectures in the future may fix the root complex issue, which will enable our software DMA engine to replace cudaMemcpy with CPU-side MMIO, or even more efficiently, DMA engines on CPU (e.g., Intel I/OAT [6] or AMD PTDMA [8]).

ACE [21] proposes offloading the entire collective communication logic to a hardware accelerator that resides on intra-machine fabric, which cannot be extended to an external network (Ethernet, Infiniband, etc). Our work differs from ACE as it is generally applicable to any (R)DMA networking and we can reuse most of existing software logic in popular collective communication libraries.

## VI. CONCLUSION

This paper envisions a GPU-driven code execution system that enables autonomous control of GPU throughout the entire lifetime of DL applications. We present the GPU-controlled DMA engine at the heart of the GPU-driven system that enables GPUs to communicate with each other without any external control. To avoid interference between computation and communication, we design our DMA engine to consume little GPU resources, so that its high communication performance is delivered without sacrificing computational throughput of GPU. While our software engine already shows benefits over commodity hardware, we also present a proof-of-concept of a hardware engine that shows even higher performance, which indicates that our system performance would be further improved with future advances in commodity hardware such as CPU, GPU, or SmartNIC.

## ACKNOWLEDGEMENTS

We appreciate the feedback by anonymous reviewers of MLArchSys'22. This work is in part supported by the ICT Research and Development Program of MSIP/IITP under projects [2022-0-00531, Development of in-network computing techniques for efficient execution of AI applications].

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
