# OpenReview forum: "Towards GPU-driven Code Execution for Distributed Deep Learning"
_iscaconf.org/ISCA/2022/Workshop/MLArchSys — MLArchSys 2022_

### Official Review · Reviewer_7Twm · 2022-05-20
**The paper argues that the current CPU-controlled and GPU-controlled using MMIO approaches face deficiency in supporting distributed training and proposes a GPU-controlled DMA mechanism to address that.**

**Rating:** 5
**Confidence:** 3

**Review:**

The paper argues that the current CPU-controlled and GPU-controlled using MMIO approaches face deficiency in supporting distributed training and proposes a GPU-controlled DMA mechanism to address that.

The research direction of the paper seems interesting and the paper provides a nice background on the current technologies. However, the manuscript requires significant improvement in terms of (1) explaining the ideas and the design, (2) evaluation, and (3) comparison with prior works.

First, the proposed DMA engine design needs better clarification and explanation (e.g using diagrams). With the current explanation the design and its novelty is not clear. Second, the paper requires more thorough study and evaluation of the design. Specifically it requires more sensitivity studies and the impact of the design on end-to-end deep learning workload execution. Moreover, the experimental setup in Figure 5 requires clarification as the numbers for the baseline Megatron-LM do not seem to be correlated with the original Megatron LM paper. The original paper shows almost linear scaling with the number of devices and decent overlap of communication and computation in data parallelism, while Figure 5 shows very poor scaling and also implies strong correlation of overall throughput with communication in data parallelism. This might be due to some details in the experimental setup, but requires more thorough explanation. Additionally, from in Section 3.B it is mentioned that the prototype only supports communication between two GPUs. However, Figure 5 shows using the proposed method for more than two GPUs. This also requires more clarification on how the communication between 8 GPUs is achieved. Lastly, it will be nice if the paper compares with the following related work:
- Enabling Compute-Communication Overlap in Distributed Deep Learning Training Platforms, ISCA 2021.

---

### Official Review · Reviewer_z4Xh · 2022-05-21
**This work identifies the inefficiency of collective communication among multi GPUs through CPU and proposes a GPU-controlled hardware DMA engine.**

**Rating:** 7
**Confidence:** 3

**Review:**

### Summary
The paper identifies the GPU-to-GPU collective communication as one of the performance bottlenecks in distributed deep learning applications. One root cause is that the existing GPU communication is CPU-based, and the communication event handling through the CPU is inefficient for collective communication among GPUs. To tackle this issue, this work suggests a GPU-centric code execution flow with GPU-thread-controlled hardware DMA added to avoid CPU intervention.

Three places could incur overhead in a CPU-controlled flow. 1. the event interface (like in Tensorflow) uses only one thread to poll multiple tasks from multi GPUs. 2. it takes a long time to wake up the CPU that calls the callback function, especially when on different cores. 3. the callback command could be delivered to GPU directly before notifying the CPU. Some inefficiency is also identified on the GPU runtime side for waking up a dependent GPU command.

Existing GPU-controlled communication (implemented in NCCL) creates an MMIO address space and uses GPU threads to perform copies to the space directly. Such implementation could use a lot of GPU resources and severely impact the performance of concurrent compute tasks (L2 cache/warp scheduler contention leads up to 2.35x/2x slowdown respectively, compute throughput dropped by 40% when running all-reduce when training BERT on 8xV100).

This work proposes to use a GPU-controlled DMA engine as a solution and discusses how to efficiently handle send request (SR) in HW and SW. Instead of coordinating with the DMA before the request, the GPU only checks a local ring buffer that the DMA asynchronously updates in this design . One the software system side, the whole deep learning application is compiled to a single loop kernel that runs repeatedly. Their results show that their software and hardware engine (G-Drv-S, G-Drv-H) significantly reduce the collective communication latency. The hardware engine can further improve throughput by handling multiple requests in parallel. End-to-end training of BERT shows a 1.82x higher all-reduce throughput improvement over two other CPU-based implementations.

Pros:
1. The paper clearly explains the key sources of inefficiency in GPU-GPU communication and quantifies the corresponding latency overhead in each activity. It provides very valuable information for both software and hardware optimization.
2. The proposal to enable a GPU-driven execution system to handle GPU-GPU communications makes a lot of sense especially for the deep learning training application. Since most of the computation and communication are on GPUs, GPU-driven execution should incur the least overhead. Meanwhile, adding DMA helps to avoid using GPU resources for communication.
3. The experimental results show significant improvement when adopting GPU-controlled communication.
4. Overall, the clarity of the paper is very good and the observations are backed by profiling data.

Cons:
1. The hardware DMA design could be better explained. e.g. It is a bit unclear to me how adding several send/recv buffers can help reduce the address space and pipelining the DMA requests. It would be great to expand this in more details.
It would be also more informative if the writer can include the resource utilization (LUTs, FFs, BRAM, DSPs) for implementing the hardware DMA engine, and if possible some energy number.
2. The implementation of loop kernel seems a little ad hoc. Does it assume a fixed execution graph that can be statically analyzed for the DL application?

Other comments:
- It seems Fig 3a) and Section IIC first part introducing CPU-controlled communication could mentioned and merged in sectionIIB .
- The abstract mentions that it is not able to control GPU's internal DMA with GPU threads, this feature might be now enabled in the latest Hopper architecture, please double check if this piece of information is still correct.

---

### Official Review · Reviewer_KiCn · 2022-05-21
**The paper is well-written. The current sets of evaluations are solid. However, more evaluations are needed to support the paper.**

**Rating:** 7
**Confidence:** 2

**Review:**

Summary:
The authors identified the latency challenge when executing parallelism in SOTA large-scaled DNN models. The authors propose a GPU-driven DMA method to communicate data between GPUs, saving latency and increasing throughput.

Pro:
* The authors base their arguments on real-system evaluation results. The data-driven discussions are solid and convincing.
* The paper is easy to follow, the structure is clear, and the explanation is well written.
* I appreciate the authors presenting two kinds of techniques, G-Drv-S and G-Drv-H. While the actual hardware support is not available (since the current GPUs do not support these types of DMA), the authors not only propose a prototyping HW implementation with FPGA but provide an intermediate SW solution to demonstrate the potential benefit. It makes the evaluation solid.

Con:
* The paper is motivated by the parallelism of DNNs. However, the evaluation of DNN models is limited. The author only demonstrates one DNN model, BERT-Large.
* In addition, the authors highlight the use case of all-to-all expert-parallelism. However, they didn’t evaluate expert-parallelism DNN models in the evaluation, leading to lack of consistency in the text and evaluation.
* Lastly for the evaluation, while the setup of the system is clearly described for Sec-IV-A, the setup for Sec-IV-B is unclear. Also, Sec-IV-B needs more illustration and discussion.
* The novelty part of the paper needs to be stated more clearly.
* There are many acronyms in the paper. It would be better to show the full terms the first time the acronyms are mentioned.

---

### Decision · Program_Chairs · 2022-05-30

**Decision:**

Accept

**Comment:**

The reviewers found the paper interesting that addresses an important problem, the GPU-to-GPU collective communication. The reviewers had some concerns about clear description of contributions and better explanation of the results. One of the reviewer suggested to provide more detailed results, such as FPGA resource utilization and power numbers. I believe that adding these results would provide more insights about the benefits and limitations of the proposed technique.

Because of the positive reviews and the ratings, the paper was nominated for `Best Paper Award`.